# Family Functioning as an Explanatory Factor of Empathic Behavior in Argentine Medical Students

**DOI:** 10.3390/bs13050356

**Published:** 2023-04-24

**Authors:** María J. Ulloque, Silvina Villalba, Gabriela Foscarini, Susana Quinteros, Aracelis Calzadilla-Núñez, Alejandro Reyes-Reyes, Víctor Díaz-Narváez

**Affiliations:** 1Facultad de Ciencias de la Salud, Universidad Católica de Córdoba, Córdoba 5004, Argentina; 2Facultad de Salud, Universidad Bernardo OHiggins, Santiago 8320000, Chile; 3Facultad de Ciencias Sociales y Comunicaciones, Universidad Santo Tomás, Concepción 8320000, Chile; 4Facultad de Odontología, Universidad Andres Bello, Santiago 8320000, Chile

**Keywords:** empathy, ex post facto study, family functioning, medicine students, resilience

## Abstract

Empathy is a relevant competence in the study and practice of medicine whose development could depend on the functioning style of each family. This study aims to compare the distribution of empathy levels, about functionality or dysfunction, and the three styles, which can be derived from family functioning in the families of Argentine medical students. Previously providing evidence of the validity of the family functioning measure. As well as provide evidence of the validity of the measure of family functioning. Methods: Ex post facto design: 306 Argentine medical students who had already taken the Jefferson Scale of Empathy—Spanish Edition (JSE-S) and the abbreviated Spanish Family Adaptability and Cohesion Evaluation Scale (FACES-20). A gender-weighted linear regression analysis was made, establishing an ANOVA and multiple comparisons via DMS to determine the effect of functional and dysfunctional families’ balanced, intermediate and extreme functioning styles concerning empathy. Results: Students who presented dysfunction in familial cohesion and adaptability showed measures of empathy greater than those classified as functional. Differences of cohesion were statistically significant in compassionate care, perspective taking and general empathy. These components were significantly higher in students from families classified as extreme than balanced ones. Students classified within families with either extreme or dysfunctional styles showed greater levels of empathy than more adaptive and functional ones, except in the ‘walking in patient’s shoes’ component where differences were not observed. Conclusions: Individual resilience as an intervening variable in the presence of empathy is discussed. Implications: The study of empathy, its associated variables, and the conditions of its development remains a central theme in relation to students and professionals of the health sciences. To achieve an effective professional practice, it is necessary to develop human capacities such as empathy and personal resilience.

## 1. Introduction

Empathy is here considered a capacity that permits not only the identification and comprehension of others’ thoughts and feelings, but emotionally appropriate responses to them [1]. Medical empathy, as relates to patient care, is conceptualized as a cognitive attribute which permits understanding of patients’ internal experiences and perspectives, and the ability to communicate that understanding [2]. Nevertheless, empathy also possesses an affective component which in clinical practice occasionally lacks depth or sincerity. Medical students have, in certain situations, felt pressure to make empathic declarations which they felt did not relate to their patient, thus experiencing discomfort and empathic dissonance due to disconnection between their statements and feelings. This is due to training that promotes the forcing and, at times, falsification of empathic declarations and time pressures which promote cognitive over affective empathy [3].

Empathy is associated with prosocial behaviors of respect and positive attitudes towards the elderly, as well as improved doctor/patient satisfaction, therapeutic relations, clinical results and the ability to compile clinical histories and make physical exams [4]. The development or academic training of empathy is relevant, paying attention to the curriculum and teaching-learning methodologies in university education [5]. The existence of factors that modulate empathic behavior has been postulated [5,6,7]. These factors can be stress, academic load, the teacher’s attitude towards the patient, personality and family functioning, among many other factors [5,6,7,8].

Various supporting antecedents of empathy are present in the family. Among school, peers, mass media and family, family is the single most important socializing agent in an individual’s life [6], both primarily, and as a nexus between individuals and society. Agents of socialization are responsible for the transmission of norms, values and behavior models. These may concur, but may also compete, reflecting society’s plurality of values and opinions rather than an undifferentiated unity [7]. Families may seek a prosocial adjustment of children via extrinsic motivators (e.g., punishment, rewards), but mostly do so via internalization of prosocial values whose intrinsic motivations guide behavior in the absence of extrinsic ones [7].

The family itself presents unique phenomena and characteristic processes. Emotional and affective bonds of family interaction make cohesion possible. It is capable of adaptably changing its own power structure, roles, rules and relationships to overcome situational stresses and obstacles to its development [9,10]. Families’ configurations change constantly, growing and adapting their structures and dynamics [11], these structural changes result in cohesion and adaptability.

Family functioning is a construct that has been extensively studied in relation to its incidence in the development and maintenance of mental health indicators in people [12,13]. Abnormal family functioning has been shown to be linked to emotional disturbances such as anxiety and depression [14,15,16]. On the other hand, adequate family functioning can contribute to a greater adaptation in young people [17,18] and a lower presence of psychological alterations [19].

As a consequence, family functioning has the potential to be a modulator of psychological wellbeing for university students and should be a matter of concern for university authorities, since it participates in the adaptive process and coping with the demands of academic activity [20]. Studies carried out on students from different countries and different health science specialties in general have observed that adequate family functioning is associated with a lower presence of symptoms of depression and anxiety [21], a decrease in the risks of anguish and stress [22,23] and would be a protective factor against risky behaviors in students [24]. On the contrary, negative family functioning is associated with a higher level of depression [25], it constitutes a risk factor for psychological distress [20] and it could be considered a poor predictor of academic performance [26].

The studies of family functioning and empathy require prior psychometric studies to verify that the model of both constructs (and their respective dimensions) is fulfilled and will thus be able to make good estimates. In relation to the ideas expressed above, one of the most used scales to study family functioning is the FACES, of which different versions of the scale have been developed. Among them are the FACES-20 ESP and FACES III scales, which evaluate linear family functioning based on the circumplex model [11,27,28,29].

Family functioning, according to Olson [10], has three dimensions: cohesion (affective unity), flexibility (to the world’s rules and of the family’s leadership) and communication (as a facilitator). This model classifies eight family categories: chaotic, flexible, structured, rigid, enmeshed, connected, separate, or disengaged [10]. Balanced and intermediate family functioning are achieved when all its goals and functions are successful; extreme functioning when these are maladaptive and less functional [10].

Dispositional and situational empathy predict altruistic and prosocial behavioral capacities and the types of parents’ discipline. Family and peer relations relate to prosocial behavior and empathy, and prosocial behavior to observable empathy [30]. Medical residents of various specialties have shown a moderate association (*r* = 0.456) between family functioning and empathy [31].

The study aims to compare the distribution of empathy levels according to family functioning and the three styles that can be derived from family functioning in the families of Argentine medical students. As well as provide evidence of the validity of the measure of family functioning.

## 2. Materials and Methods

### 2.1. Participants

The population was made up of 497 medical students; a sample of 306 was analyzed (61.57% of the total). The sample stratified by school year was as follows: first year = 49, second year = 40, third year = 95, fourth year = 46 and fifth year = 76. In terms of sex, the sample was distributed into 195 female students and 111 male students. The total student population was not analyzed because (a) they were absent or late for class (125 students) on the day the scales were administered, (b) the instruments were not fully completed (36 students) or (c) they did not agree to participate voluntarily (30 students). The scale was not administered to students who were absent to prevent any potential answer contamination. Therefore, the sample was not chosen randomly and can be considered a convenience sample [32].

### 2.2. Instruments

The Jefferson Scale of Empathy for medical students, Spanish edition (JSE-S) was used, adapted for Argentine use by a panel of judges. It contains 20 items across 3 dimensions: 10 for perspective taking, 8 for compassionate care and 2 for walking in the patient’s shoes. Each is a Likert scale of 1–7 (strongly disagree—strongly agree), giving a score of between 20 and 140 for the whole test and maximums of 70, 56 and 14 for each area. The instrument’s psychometry is valid, reliable and exhaustively studied among medical, nursing and dental students [2,33,34,35,36,37]. In all the aforementioned studies, it was possible to verify the construct validity of the JSE-S instrument, consisting of a construct with three latent dimensions: compassionate care, perspective taking and “walking in the patient’s shoes” with 8, 10 and 2 items, respectively.

The abbreviated 20-item Spanish Family Adaptability and Cohesion Evaluation Scales (FACES-20 Esp.) were used to evaluate family functioning. Each item describes a family situation with a Likert scale of 0–4 (never—almost always). This version has been validated in Spain [38], Chile [11] and Ecuador [7]. Two dimensions, cohesion and adaptability, as revealed by exploratory factor analysis in Chile and Spain and confirmatory factor analysis in Spain [11,38], proved consistent with Olson’s circumplex model [10]. Each dimension is split into four levels from low to high: disengaged, separated, connected and enmeshed for cohesion; rigid, structured, flexible and chaotic for adaptability. From the combinations of the levels of cohesion and adaptability, the following family typology is extracted: balanced, intermediate, and extreme [11]. The psychometric properties of this instrument were evaluated before making group comparisons. As a result of this procedure, a FACES measure based on 15 items is validated and used (FACES-15), eliminating items 4, 9, 12, 14, and 20. The study of the psychometrics of family functioning was necessary because there is a methodological requirement to verify compliance with the model in populations that are little studied [39].

### 2.3. Procedures

The data were collected in November 2019. The scales were applied in paper format, then they were tabulated in an Excel table designed to contain information about the data of the academic year in which they were, sex, age, empathy data and family functioning.

#### Bioethical Considerations:

The study was carried out in compliance with the regulations of the Declaration of Helsinki, Good Clinical Practices of ANMAT and Provincial Law No. 9694. Protection of the personal data of patients is ensured according to Law 25326. Ethical approval was obtained from the Institutional Committee of Ethics in Health Research dated 30 June 2019.

Before responding to the instruments applied, the students had to sign an informed consent and had the right to bring a copy of it in which they were informed of their ethical rights, including the possibility of waiving their data to be processed. The absence of a document signed by the participant was implied as a criterion for exclusion from the investigation. The data of the participants were confidential.

### 2.4. Data Analysis

The Kolmogorov–Smirnov test was used to determine the univariate normality of the empathy data and family functioning, and Mardia’s [40] multivariable asymmetry and kurtosis analysis was used to study multivariable normality. We used Levene’s test to study the homoscedasticity of the variables. The internal consistency of the measurements was estimated using Cronbach’s alpha coefficient [41] and the concordance between the intraclass correlation coefficient. Standard measurements and deviation samples were measured; using the Mann–Whitney U test to compare measures of empathy by students classified as being in functional or dysfunctional families by either cohesion or adaptation. Cohen’s d was calculated as a measure of effect size. ANOVA with least-squares regression by gender was also used to determine if this factor would reveal typological differences in family functioning styles. Partial eta squared was used to determine the effect’s size (*η_p_^2^*). The data were described via descriptive statistics. Both exploratory (EFA) and confirmatory (CFA) factor analyses were used to analyze the factorial structure of the FACES-20 Esp., together with a generalized least squares (GLS) estimation based on the covariance matrix. The following were used to evaluate the fitness of the models: (a) a Chi–squared test (χ^2^); (b) a normalized Chi–squared test (χ^2^/df); (c) goodness (GFI), comparative (CFI) and adjusted goodness of fit (AGFI) indices; (d) root mean square error of approximation (RMSEA) and (e) root mean square residuals (RMSR). Adequate fitness values were, for (a): non-significant, (b) <2, (c) ≥0.90 as acceptable, ≥0.95 as good, (d) ≤0.05 (90% CI ≤ 0.08) and (e) 0.08 as acceptable and 0.06 as excellent [18,19,20]. Factorial loads ≥0.40 were considered significant [42]. The level of significance was α < 0.05 and β < 0.20 in all cases, and the analyses were carried out by IBM SPSS Statistics 25 (IBM Corp., Armonk, NY, USA); Factor Analysis 12.03.01 (Lorenzo-Seva, U. and Ferrando, P. J., Tarragona, Spain) and Amos 25 (Arbuckle, J. L., IBM Corp., Chicago, IL, USA).

## 3. Results

### 3.1. Psychometric Analysis of FACES

Given the model’s lack of evidence regarding family functioning in Argentine medical students, we performed an EFA using a random partition of 40% of the sample´s data. From this, a significant Bartlett’s result (Bartlett’s Statistics = 3436.6, df = 190, *p* < 0.0001) was obtained, with a Kaiser–Meyer–Olkin (KMO) test result of 0.88, CI 95% (0.877; 0.880), the polychoric correlation matrix was determined to be adequate for EFA as the univariate distribution of ordinal elements were asymmetric or had an excess of kurtosis. These items did not present normal, univariate distribution (i.e., all values *p* < 0.001 according to the Kolmogorov–Smirnov test) and, consequently, lacked normal multivariate distribution: Mardia’s multivariate kurtosis = 588.06, *p* < 0.00001. Parallel analysis (PA) based on minimum rank factor analysis was used to determine the two factors that coincided with the literature: family cohesion (items 1, 5, 8, 10, 11, 13, 15, 17 and 19) and adaptability (items 2, 3, 6, 7, 16 and 18), explaining 58.83% of the total variance. Five items (4, 9, 12, 14, and 20) showed factorial loads contrary to those specified in the original model and were not theoretically pertinent as they do not measure these dimensions. We called this abbreviated model “FACES -15”.

The remaining 60% of the sample was analyzed with CFA and GLS to find factorial validity, confirm the structure of latent variables and test a three-model theoretical structure: (a) the original Spanish model of twenty items divided evenly between each factor [15], (b) the original incorporating correlated errors (items 1–4, 1–7, 2–6, 3–16, 5–13, 8–15, 8–13, 11–13, 11–15, 7–9, 9–12, 14–16, 6–18) and (c) the fifteen question model mentioned above with correlated errors between items 6–18, 5–11, 5–13 and 8–13 (see Appendix A (Appendix A), data and Appendix A are openly available and can be accessed at https://osf.io/pjfyu/ accessed on 3 March 2023). The goodness of fit indices are presented in Table 1, showing the best fit for model c (GFI = 0.921, RMSEA = 0.06), as well as adequate and significant standardized factorial loads, varying between λ = 0.42 and λ = 0.82.

### 3.2. Measurement Reliability

Reliability was established using standardized Cronbach’s alpha (α *), non-standardized Cronbach’s alpha (α) and the intraclass correlation coefficient (ICC). Adequate values were observed for the FACES-15: α * = 0.87, α= 0.86 (CI 95% = 0.85, 0.86), ICC = 0.87 (*p* = 0.001) and for the JSE-S: α * = 0.79, α = 0.77 (CI 95% = 0.74, 0.78), ICC = 0.77 (*p* = 0.001).

### 3.3. Empathy by Functionality

Table 2 shows measured empathy values in families and includes each of the dimensions of empathy and family functioning. Classified for cohesion, 43.8% were functional and 56.2% were dysfunctional; for adaptability, 77.1% and 22.9%, respectively.

After the application of the Mann–Whitney U test, greater empathy was observed in families classified as dysfunctional when sorted by cohesion, greater compassionate care (*p* < 0.01), perspective taking (*p* < 0.05), and overall empathy (*p* < 0.01) than those students who perceive good family functioning. For adaptability, no statistically significant differences were observed between groups; however, slightly higher scores in all components were observed in dysfunctional families (Table 2).

### 3.4. Empathy by Family Functioning Style

Table 3 shows the results of a gender-weighted least squares regression estimation of empathy measurements for each family functioning style. Extreme families presented the highest empathy measurements, with significant differences between groups except in the walking in the patient’s shoes component. DMS testing revealed higher overall empathy (*p* = 0.005), compassionate care (*p* = 0.017) and perspective taking (*p* = 0.018) in extreme cases than balanced ones (*p* = 0.029), with no difference in other components.

## 4. Discussion

The fact of having proven the existence of an agreement between the models of the family functioning and empathy construct (and their underlying dimensions) with the observed data is not a trivial problem. This compliance, together with the estimation of the invariance of the model between groups and the values of internal consistency, makes it possible to distinguish that the items that are located in each latent dimension of the studied constructs are located in the corresponding dimensions, that the comparisons between groups (regardless of their nature) can be performed with confidence based on the variables of interest and, finally, if the internal consistency is adequate, the error of a bad decision is avoided. In the present work, the multi-group factorial analysis was not performed because it was not an objective of the present study.

In the case of empathy, the magnitude achieved in its three dimensions can be quantitatively evaluated. Therefore, the closer we get to the maximum value that the items corresponding to each dimension can reach, we could infer that the levels of empathy are higher and, therefore, it can be decided if one person can have more or less empathy than the other. In the case of family functioning, the correct distribution of the items in each of the two dimensions that this instrument has, cohesion and adaptability, will make it possible to distinguish (according to the value reached in the corresponding items). Cohesion dimension (family of four forms): disconnected, separated, connected and agglutinated. Adaptability dimension (four familiar shapes can be inferred): rigid, structured, flexible and chaotic. Among the family forms, within cohesion and adaptability, dysfunctional and functional families can be obtained, and the combinations of these family forms will allow families to be classified by family typologies or styles of family functioning. These typologies are balanced, intermediate, and extreme families. A question possibly arises from this situation: can we carry out association studies between two constructs, such as empathy and family functioning, without first verifying that the observed data, using the relevant instruments, fit the corresponding model? The authors of this study are inclined to answer that it is a risk to assume compliance with a model without extensive empirical evidence that, in the case of the family construct, is very scarce in Latin America.

Combining measured perceptions of family functioning levels (low to high adaptability and cohesion) and styles (balanced, intermediate, and extreme) according to Olson [10,11,38] showed slightly higher overall empathy values in students from families classified as dysfunctional. However, statistically significant results were found only in cohesively dysfunctional families; their students had higher scores in overall empathy, compassionate care, and perspective taking. In the literature, we only found one work that specifically tried to find some kind of relationship between empathy and family functioning [7]. The results of this work described that the highest values of empathy were also characteristic of the students who referred to their families as extreme.

These results could be explained, theoretically and in part, by means of a working hypothesis (of an exploratory nature) [43] consisting of the fact that resilience could be a factor (among many others) [2,4,5,6,7,8,32] that would contribute to the positive development of empathy in a relatively negative family context. Indeed, resilience could be characterized, in summary, by a high degree of expectation, self-determination, flexibility, optimism, cognitive reappraisal, and active coping [44,45,46,47]. Individual resilience can be considered a contextual ability through which people can manifest resilience in certain and determined circumstances, but not in others [48]. In synthesis, subjective and psychosocial factors can strongly influence individual resilience [49]; the same factor can generate differentiated effects in accordance with the person’s specific cognitive and emotional capacities and social resources. Resilience can only manifest itself after a negative event and consists of coping with the said event before achieving individual balance. Said balance is only possible through a process of self-education that allows acquiring new resilient characteristics or strengthening existing ones or both at the same time [50].

Resilience can also be analyzed from the evolutionary point of view. The possibility of maintaining allostatic equilibrium allows us to suggest that evolution should be in favor of resilient people [51].

In the neurobiological substrate of resilience, there are neurological structures that are involved in the various systems related to the capacity to respond to stress and individual resilience [52]. The hypothalamic-pituitary-adrenal (HPA) axis is a neuroendocrine system that is involved in an organism’s adaptation to stressful events [49]. In addition, there are other neural mechanisms involved in resilience: the locus coeruleus/noradrenaline system, the mesolimbic reward circuit and the fear circuit. There are several psychic aspects associated with fear (learning fear, memory, responses, modulation and extinction). These aspects are regulated and associated with brain areas including the amygdala, hippocampus, medial prefrontal cortex (MPC), nucleus accumbens (NAc), ventromedial hypothalamus and brainstem nuclei, all of which are involved in brain control of related processes under stress conditions [45,47].

The existence of resilient phenotypes has been proposed, which constitute the expression of genes determined and specific polymorphisms that regulate the functioning of the HPA axis, neuropeptide Y, and the noradrenergic, dopaminergic, and serotonergic systems [47] that, together with the autonomic nervous system, increase the likelihood of vulnerability to depression and other psychiatric disorders [52]. Consequently, resilience is not limited to the psychological and behavioral adaptation to a stressful situation, it also implies the functional neurobiological reaction to said situation. This characterization of resilience allows us to infer that it is a complex system where there is a very strong relationship between the psychological architecture of a person, the physiological reaction to a stressful event and the event itself.

Psychological resilience has been associated with changes in the left orbitofrontal cortex. This is an area of the brain that belongs to the neural circuits that govern various processes, but among them is the regulation of emotions [53].

Positive emotions, particularly optimism and humor, are important psychosocial factors of resilience [46]. It has been suggested that the expectation of having the optimism of a good future has a 25% genetic load, but, in addition, it can be increased through specific psychological interventions [54]. On the other hand, cognitive flexibility is also a mechanism for gaining resilience and includes cognitive reappraisal; the ability to reframe experiences from a more positive perspective [55] increases resilience. Resilient people are also characterized by their moral structure and the practice of religious/spiritual beliefs and altruism [55]. The altruist can give a passive meaning to his life and its context in the presence of an adverse event; therefore, altruism contributes to resilience.

Social support is another factor that stimulates resilience. A systematic review consisting of 36 works [56] showed that 89% of them observed a significant association between social support and protection against depression in adults.

However, this work cannot explain exactly how resilience is associated with empathy. Both constructs are the theoretical reflection of complex neuroanatomical systems that possibly developed in parallel in phylogenetic and ontogenetic processes, and ontogeny could also be very determinant in the formation of resilience, just as it occurs with empathy [5,6,7,8,32].

Individual and community resilience develops in conditions of adversity [57], such as may be found in dysfunctional and extreme families. The ability to meet challenges, together with a prosocial attitude are useful tools for empathy [58]. Resilience allows us to empathize with those who have passed through difficult times, as seen in the significant association between resilience and empathy among those with sick partners [59]. Individual resilience depends upon affective support and individual strength in vulnerable situations. On the other hand, family resilience implies a salute genic and positive perspective in that it explains why some families show good adaptation despite being exposed to severe adversity [59].

Empathy, combined with resilience, promotes prosocial interactions, reciprocity, and the experience of positive emotions in oneself and others. Empathy can empower people to see interpersonal relationships as opportunities for learning and growth; resilience allows them to enrich their experiences, see them from various perspectives [60,61] and retain the emotional and mental flexibility necessary to interpret life’s difficult situations as challenges and opportunities rather than obstacles and threats.

The study of empathy, related variables and conditions of its development remains a topic of interest for students and professionals of health sciences. Indeed, several authors have carried out empathy studies in Latin America in students of Health Sciences: nursing, medicine, dentistry, kinesiology, and nutrition [5,6,7,8,32,62,63,64,65,66,67,68,69,70,71,72] and have shown that the distribution of empathy throughout the courses and in the sex is variable. Specifically, that the premise of the existence of empathic decline and that the levels of empathy between the sexes always favor women, is not fulfilled in some studies [62,63,64,65,66,67,68,69,70,71,72]. There are also works that associate the levels of empathy with personality in medical students [8], with teamwork, permanent learning, and family loneliness, also in medical students [73] and that individual characteristics and those related to society and the family are related to the development of empathy [74]. In general, it is possible to affirm that empathy is a product of several factors that act at the same time on the complex processes involved in the empathic formation of a student, including culture [75].

Such variability can only be explained by the presence of endogenous and exogenous factors [76] that act on people at all stages of their development [65]. This similar situation occurs with resilience in relation to the process of operationalizing resilience [77].

Effective professional practice requires empowering students’ human capacities and personal resilience. A challenge not only for communicative competencies but for all affective and relational competencies which contribute to better health services for communities.

Being an ex post facto study, it is impossible to establish causal relationships between family functioning and empathy. This would require a longitudinal study. We consider this present study a base from which to project new, deeper studies into the same area.

To achieve a resulting effective professional practice, it is necessary to enhance the building of human capacities and personal resilience. In this sense, it is a challenge to improve not only communication skills but also all those affective and relational skills, such as empathy, and apply them to provide better health services for communities.

## 5. Conclusions

In conclusion, measuring family cohesion and adaptability to determine functionality showed that students from dysfunctional (disengaged, enmeshed, rigid or chaotic) families scored more highly in general empathy, compassionate care, and perspective-taking elements than those from functional (separated, connected, structured or flexible) ones. Only in walking in the patient’s shoes were no differences between these groups observed. Balanced and extreme families scored similar empathy results, with extreme ones scoring higher than average balanced ones’ scores in the same empathy components, except in walking in the patient’s shoes. Intermediate families scored more highly than balanced ones in compassionate care, with no other statistically significant difference between family functioning styles.

## Figures and Tables

**Table 1 behavsci-13-00356-t001:** FACES’ CFA goodness of fit indices.

Models	χ^2^	*p*	χ^2^/df	RMSR	GFI	CFI	RMSEA [CI 90%]
Model a	462.220	0.0001	2.735	0.096	0.848	0.287	0.075 [0.067; 0.084]
Model b	342.434	0.0001	2.195	0.084	0.888	0.547	0.063 [0.054; 0.072]
Model c	179.767	0.0001	2.115	0.064	0.921	0.687	0.060 [0.048; 0.073]

Note: GFI = goodness of fit index, RMSR = root mean square residuals, CFI = comparative fit index, RMSEA = root mean square error of approximation, CI = confidence interval. Model a = original model of 20 items, Model b = model with correlational errors, Model c = FACES–15.

**Table 2 behavsci-13-00356-t002:** Comparison of empathy measurements in students from functional and dysfunctional families.

	Family Functioning								
	Functional		Dysfunctional						
Dimension	M	SD	M	SD	Z	*p*	d	CI 95% Diff.	
Cohesion									
Overall empathy	110.88	10.94	114.36	11.86	−2.84	0.004	0.305	−6.08	−0.88
Compassionate care	45.04	6.47	46.95	5.95	−2.57	0.010	0.307	−3.30	−0.50
Perspective taking	58.27	6.85	59.92	7.27	−2.39	0.017	0.234	−3.26	−0.04
Walking in patient’s shoes	7.57	2.23	7.49	2.67	−0.26	0.799	0.033	−0.48	0.63
Adaptability									
Overall empathy	112.19	11.57	115.01	11.40	−1.79	0.074	0.246	−5.91	0.27
Compassionate care	45.81	6.23	47.13	6.23	−1.89	0.058	0.212	−2.98	0.35
Perspective taking	58.88	7.11	60.26	7.12	−1.50	0.133	0.194	−3.30	0.54
Walking in patient’s shoes	7.50	2.45	7.63	2.60	−0.72	0.475	0.051	−0.83	0.56

Note: SD = Standard deviation, M = Mean, Z = Mann–Whitney’s U, *p* = *p*-value, d = Cohen´s d, CI = Confidence interval for the difference between means.

**Table 3 behavsci-13-00356-t003:** Comparison of empathy by family functioning styles.

Component	Functioning Style	N	M	SD	F	*p*	$\eta_{p}^{2}\;$
Overall empathy	Balanced	115	110.72	11.01	5.869	0.001	0.055
	Intermediate	140	113.35	11.78			
	Extreme	51	116.20	11.52			
	Total	306	112.84	11.58			
Compassionate care	Balanced	115	44.79	6.51	5.747	0.001	0.054
	Intermediate	140	46.76	5.83			
	Extreme	51	47.33	6.31			
	Total	306	46.11	6.24			
Perspective taking	Balanced	115	58.38	6.79	3.232	0.023	0.031
	Intermediate	140	59.11	7.40			
	Extreme	51	61.25	6.84			
	Total	306	59.20	7.13			
Walking in the patient’s shoes	Balanced	115	7.55	2.22	0.319	0.812	0.003
	Intermediate	140	7.48	2.62			
	Extreme	51	7.61	2.71			
	Total	306	7.53	2.48			

Note: N = Sample size, M = Mean, SD = Standard deviation, F = Fisher-Snedecor distribution, *p* = *p*-value, $\eta_{p}^{2}$ = partial eta squared (effect size).

## Data Availability

Data is openly available and can be accessed at https://osf.io/tk9d6.

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
