# Peer review of "Family Functioning as an Explanatory Factor of Empathic Behavior in Argentine Medical Students"

_behavsci, 2023, doi:10.3390/bs13050356_

Round 1

Reviewer 1 Report

Dear Authors,

The theme of the work carried out is pertinent and falls within the scope of the International Journal of Environmental Research and Public Health.

The research is very important as it discusses the influence of the family on development of empathy in students.

As a reviewer, I provide the following suggestions to maximize the strength of the article:

Introduction

However, this section is lacking a consistent overview of completed research in this area and what the author’s work can add to this area of study.

Participants

Please complete the basic description of the study participants, e.g. age, gender (and possibly what is their status at University such as freshman, sophomore, junior, or senior).

Discussion

Interesting results and an interesting interpretation of research results showing higher empathy in students from dysfunctional families. I would recommend looking for other studies, the results of which will confirm such a relationship or contradict it.

Author Response

Dear Colleague Reviewer 1:

Before responding to your suggestions, I want to thank you for your review because the inclusion of these suggestions in the text has undoubtedly entichrd the work. I hace incluided all your suggestions hoping that you can satisfy them. All of them are in red color in the text.

Introduction: We have expanded the Introduction and we estimete that the inclusions made can provide a general vision of the theme of this work.

Participants: We have completed the basic description of the study participants.

Discussion: We have included investigations about the possible factors that determine the development of resilience and the same procedure has been carried out with empathy. But we have not found works that associate empathy with resilience and, even less, works that report negative or positive about this association.

Sincerely,

Reviewer 2 Report

I read the manuscript by María J. Ulloque et al. entitled "Family functioning as an explanatory factor of empathic behavior in Argentine medical students". The topic is interesting and original. This study could be of high quality. However, the authors at the end of the introduction decided to shoot themselves in the foot.

Without any previous reference in the title or in the introduction they add as the purpose of the study "Secondly, it evaluates the psychometric properties of the functional measurement used". Why? Completely absent from the Introduction is the reason for the factorial analysis of the questionnaire. After the factor analysis was done on the Results, we find that we do not find any comments on this analysis, neither in the discussion nor in the conclusions nor in the summary. Through the factor analysis the authors come up with a new model of Faces-15 that is better than the one the authors use in the rest of the results, this fact seriously reduces the value of the rest of the findings.

I still think that the description of the participants is missing; it is not possible to replace this with a reference to another paper, while it is not legitimate to refer to link to show the results of this paper.

In conclusion, I believe that this manuscript can be published after minor revision. The changes required are:

 -Removal from the introduction of the phrase "Secondly, it evaluates the psychometric properties of the functional measurement used".

-Removal of the results of the factor analysis (the creation of Faces-15 should be a new paper).

-Provide the description of the participants.  Add how you determined the sample size.

- Correct to "Bioethical Considerations" : ...dated June 30, 2019 to a possible ...dated June 30, 2016.

Author Response

Dear Colleague Reviewer 2.

Before responding toyour suggestions, I want to thank you fot yourb review because the inclusion of these suggestions un the text has undoubtedly enriched the work. I have included all your sugguestions hoping that you can satisfy them, All of them are in red color in the text.

  1. The suggestions to remove the following from the objective; " Secondly, evaluate the psycometric properties of the functional measure used" is absolutely reasonable and correct on your part. Consecuently, we have removed it from the text.
  2. Howevwr,although to need to carry out factorial procedures is not the objective of thos study, we have considered keeping this analysis in the text. Such consideration is not due to an intransigent position on our part. This is due to the methodological requirement consisting of reliably verifying that the data strictly conform to the models of both constructs studied and thus avoiding that the associations (positive or negative) found are devoid of errors associated with the validity of the construct, We have often reviewed articles and seen publications that try to associate two constructs without having proven its validity, based on the fact that there are several reports that indicate such a situation. The fact that a construct model has been proven to be applicable to a data set in a given population does not necessarily imply that the exact same thing happens in other populations. This is due to the heterogeneity of the populations due cultural factors among many others.
  3. Not having commented more extensively on the results of the factor analyzes was also an error on our part, as described in your critique. But, once eliminated as an objective, what corresponds to do is to describe the psichometric results, which are in the original text. But based on your suggestion, we decided to refer, in the Discussion section, to the meaning of having found fit of the models and adecuate reliability indices, but not in the conclusions since it was  eliminated as an objective.
  4. Based in your sugesstion, we have made a more detailed description of the participants.
  5. We have fixed the error regarding the ethical approval dates of the research and the start of data collection.

Sincerely,

Correspondig author
